# Population Structure of Modern Winter Wheat Accessions from Central Asia

**DOI:** 10.3390/plants12122233

**Published:** 2023-06-06

**Authors:** Akerke Amalova, Kanat Yermekbayev, Simon Griffiths, Mark Owen Winfield, Alexey Morgounov, Saule Abugalieva, Yerlan Turuspekov

**Affiliations:** 1Laboratory of Molecular Genetics, Institute of Plant Biology and Biotechnology, Almaty 050040, Kazakhstan; akerke.amalova@gmail.com (A.A.); kanat.yermekbayev@gmail.com (K.Y.); absaule@yahoo.com (S.A.); 2Crop Genetics Department, John Innes Centre, Norwich NR4 7UH, UK; simon.griffiths@jic.ac.uk; 3Faculty of Life Sciences, University of Bristol, Bristol BS8 1TQ, UK; mark.winfield@bristol.ac.uk; 4Science Department, S. Seifullin Kazakh Agrotechnical University, Astana 010011, Kazakhstan; alexey.morgounov@gmail.com; 5Faculty of Biology and Biotechnology, Al-Farabi Kazakh National University, Almaty 050040, Kazakhstan

**Keywords:** *Triticum aestivum* L., genetic diversity, population structure, single-nucleotide polymorphism, Affymetrix Axiom SNP array

## Abstract

Despite the importance of winter wheat in Central Asian countries, there are limited reports describing their diversity within this region. In this study, the population structures of 115 modern winter wheat cultivars from four Central Asian countries were compared to germplasms from six other geographic origins using 10,746 polymorphic single-nucleotide polymorphism (SNP) markers. After applying the STRUCTURE package, we found that in terms of the most optimal K steps, samples from Kazakhstan and Kyrgyzstan were grouped together with samples from Russia, while samples from Tajikistan and Uzbekistan were grouped with samples from Afghanistan. The mean value of Nei’s genetic diversity index for the germplasm from four groups from Central Asia was 0.261, which is comparable to that of the six other groups studied: Europe, Australia, the USA, Afghanistan, Turkey, and Russia. The Principal Coordinate Analysis (PCoA) showed that samples from Kyrgyzstan, Tajikistan, and Uzbekistan were close to samples from Turkey, while Kazakh accessions were located near samples from Russia. The evaluation of 10,746 SNPs in Central Asian wheat suggested that 1006 markers had opposing allele frequencies. Further assessment of the physical positions of these 1006 SNPs in the Wheat Ensembl database indicated that most of these markers are constituents of genes associated with plant stress tolerance and adaptability. Therefore, the SNP markers identified can be effectively used in regional winter wheat breeding projects for facilitating plant adaptation and stress resistance.

## 1. Introduction

Wheat is the third most important crop in the world after rice and maize and the most important in Central Asia. In 2023, the FAO pegged wheat production at 784 million tons globally [1]. In five of the former Soviet countries (Kazakhstan, Kyrgyzstan, Tajikistan, Turkmenistan, and Uzbekistan) that are geographically situated in the Central Asian region, wheat is grown in an area that exceeds 15 million hectares (ha), with production at 22 million tons. In this region, wheat is the primary food source and an essential animal feed resource [2].

Wheat is commonly classified into spring and winter types; the former requires a long period of low temperature to accomplish vernalization [3]. Genetically, these two types of wheat are determined by the expressions of three *Vrn* genes in the vernalization response process [4,5]. In Central Asia, these two types of wheat show different priorities in terms of growth, as spring wheat dominates in Kazakhstan and winter wheat prevails in the other countries. The predominance of spring wheat in Kazakhstan is determined by the prolonged cold winter in this country, during which the average temperature in Northern Kazakhstan (the major wheat-growing area) is around −20 °C [6]. However, winter wheat grows successfully in the southern and southeastern regions of the country given their relatively mild weather. Collectively, the Central Asian countries (Kazakhstan, Kyrgyzstan, Tajikistan, Turkmenistan, and Uzbekistan) grow winter wheat over a territory exceeding 3 million hectares annually [7]. However, the sowing area for winter wheat in Kazakhstan has been gradually decreasing, from 1 million ha in the early years of the century to the current 559 thousand ha [8], which is associated with the governmental policy of crop diversification and the accompanying increase in land devoted to legumes and technical crops [9]. The average yield of winter wheat in the region increased from 1.5 t/ha in 1996 to 3.1 t/ha in 2020 and varies from 1.9 t/ha in Kazakhstan to 4.5 t/ha in Uzbekistan [1]. As the average yield of winter wheat in the region is three times higher than that of the spring type in Kazakhstan, it represents a favorable wheat cultivation approach for local farmers.

Geographically, Central Asia is situated along the ancient Silk Road, and the earliest records of wheat cultivation are from Uzbekistan, dating from the second millennium BCE [10]. The region has been identified as a global genetic point of origin for this crop [11]. According to Udachin and Shahmedov (1984), the entire history of wheat breeding in Central Asia can be divided into three periods. The first covers the 18th and 19th centuries, before the establishment of the Turkestan district in 1867. The second is considered to have been relatively short, starting in 1867 and ending at the time of the Russian revolution in 1917. The third includes the period during which Central Asian countries were a part of the Soviet Union [12]. While the first and second periods have been poorly accounted for in the scientific literature, the third has been well documented by the Vavilov All-Russian Research Institute of Plant Production [12]. In the currently ongoing fourth period, following the breakup of the USSR, breeders in Central Asian countries have been closely collaborating over the last thirty years via bilateral projects and international activities pursued as part of the CIMMYT [2,13]. However, there remains little information available on the genetic structure of modern cultivars, as well as little understanding of the populational structure of Central Asian winter wheat, which is a major wheat type in this region (southern and southeastern Kazakhstan). It is assumed that the genetic structure of modern winter wheat accessions in this region was formed via long-term breeding using two distinct germplasm pools. The first germplasm pool was founded on the historical activities pursued along the Great Silk Road [14,15] and the continuing direct supply of germplasm from Western Asia. Therefore, the majority of accessions are assumed to have “genetic signatures” from Turkey and other countries from Western or Southwestern Asia, such as from neighboring Afghanistan. The second pool was formed using the germplasm from Eastern European territories and was the result of wheat cultivation activities undertaken during the Soviet era, when most of the sources used for the development of new cultivars were accessions taken from the territories of modern Ukraine and the Russian Federation [13,16,17]. Thus, it will be rewarding to evaluate the genetic diversity and population structure of contemporary winter wheat accessions in one of the largest wheat-growing areas in the world, as this may shed light on the adaptation of genotypes to specific environments. Recently, a bank comprised of winter wheat collected from Central Asia was established as part of the CAWBIN (Central Asian Wheat Breeding Initiative), launched as the collective effort of Central Asian and British scientists in Kazakhstan in 2018. As a result, modern winter wheat accessions from Kazakhstan, Kyrgyzstan, Tajikistan, and Uzbekistan, along with landraces and cultivars from Turkey, Afghanistan, and Europe, were characterized. As such, this collection could be effectively utilized for breeding purposes, by using informative genotyping systems to assess the genetic diversity and population structure and evaluate the impacts of two different germplasm pools on winter wheat breeding in the region.

In the last decade, the reliable evaluation of the genetic variation in wheat collections has become feasible due to the significant progress made in the development of efficient SNP (single-nucleotide polymorphism) genotyping platforms, established via the efforts of the international wheat community [18,19]. To date, there is a large list of wheat SNP genotyping platforms that have been successfully introduced, including 9 K [20], 15 K [21], and 90 K Illumina SNP arrays [22] and 35 K [23], 55 K, and 820 K [24] Axiom^®^ arrays. Notably, with the use of the 90 K SNP Illumina array, a set of 90 accessions of modern spring wheat from Kazakhstan were genotyped and compared with the SNP genotyping data of 690 wheat accessions representing landraces and varieties from Asia, Australia, Canada, Europe, and North America [25]. As a result, the spring wheat from Kazakhstan was grouped with samples from the USA and Europe. However, the authors did not evaluate the winter accessions from Kazakhstan and other Central Asian countries [25]. Hence, a separate study of winter wheat accessions from this relatively under-studied region would yield interesting results in assessing the population’s genetic structure and identifying critical regions of the genome responsible for intra- and interregional adaptation processes.

## 2. Results

### 2.1. Genetic Diversity and Genetic Distances in Winter Wheat Populations

Nei’s unbiased diversity index values in the studied populations ranged from 0.212 in Kazakhstan to 0.325 in the USA (Table 1). The highest genetic diversity index in the CA region (0.282) was recorded in Uzbekistan, followed by Tajikistan (0.28) and Kyrgyzstan (0.27) (Table 1). The mean genetic diversity for the CA samples was 0.261, and these values were comparable in both the region’s northern (Russia, 0.258) and southern (Afghanistan (0.259) neighbors.

Nei’s genetic distances ranged from 0.003 (between Uzbekistan and Kyrgyzstan) to 0.28 (between Afghanistan and Europe) (Table 2). Within the CA region, the largest genetic distance was between Kazakhstan and Tajikistan (0.042).

The PCoA indicated the genetic relatedness of accessions from Uzbekistan, Tajikistan, and Kyrgyzstan, all of which were positioned at a distance from the Kazakh accessions (Figure 1). The PC1 test clearly distinguished the CA accessions from the samples from Europe, Australia, and the USA, while PC2 further distinguished the accessions from Afghanistan, which were positioned in the left upper corner of the PCoA plot (Figure 1).

### 2.2. Population Structure of the Winter Wheat Accessions from Central Asia

The results of the “Evanno test” and the “elbow method” suggest that the optimal number of K steps throughout the entirety of the studied collection was 4 (Figure 2), at which point the line reached a plateau.

The STRUCTURE output at step K4 indicates that most samples from CA belonged to cluster 1 and cluster 3 (Figure 3). Interestingly, all samples from Afghanistan (100%) were grouped into cluster 1, while most of the samples of cluster 3 were from the Russian Federation (92.6%). The results show the different genetic structures present in the samples from the CA (Figure 3). The samples from cluster 1 were dominant in Tajikistan (66.7%) and Uzbekistan (51.9 %), and samples from cluster 3 prevailed in Kazakhstan (86.5%) and Kyrgyzstan (66.7%).

Evaluations of the following K steps (from K5 to K10) yielded similar trends, as samples from Tajikistan and Uzbekistan fit into clusters that they shared with samples from Afghanistan, and samples from Kazakhstan and Kyrgyzstan had patterns in common with accessions from the Russian Federation (Appendix A). Nevertheless, at step K10, both Tajikistan and Uzbekistan showed a significant number of accession groups that they shared with samples from Russia in cluster 4 (Appendix A). Interestingly, at step K10, 21 samples from Afghanistan (65.6% of the total), 4 samples from Uzbekistan (14.8%), and 2 samples from Kazakhstan (3.8%) were grouped together in cluster 10. Additionally, the assessment of samples from other parts of the world at step K10 suggested that those from the USA and Australia had similar structures of genetic diversity, while samples from Europe showed exceptionally high variation in 8 out of 10 clusters (Appendix A).

### 2.3. The Evaluation of the Winter Wheat Collection Based on Chromosomes Using Principal Coordinate Analysis

In addition to the PCoA plot of SNPs that applied to the entire genome (Figure 1), individual plots were generated for each wheat chromosome (Appendix A). The PCoA analysis for the entire genome was based on using 10,746 SNP markers, including 4129 SNPs in the A genome, 5080 SNPs in the B genome, and 1526 SNPs in the D genome. The minimum number of SNPs per chromosome was in chromosome 6D (115), while the maximum number was in chromosome 1B (975) (Appendix A). The average contributions to the total variations in PC1 and PCA 2 were 61.35% and 20.98%, respectively. The contribution made by PC1 ranged from 49.5% for chromosome 6A to 93.5% for chromosome 4D. The contribution made by PC2 ranged from 4.4% for chromosome 4D to 33.8% for chromosome 2D (Appendix A). In addition, the PCoA plots of the groups of SNP markers in individual chromosomes helped in the discrimination between samples from different countries in Central Asia. For instance, five plots (2A, 5A, 6A, 6D, and 7A) discriminated between samples from Kazakhstan and Tajikistan via PC1. Another five plots (1D, 2A, 3B, 4B, and 7B) were distinguished between samples from the same regions using PC2. The exception to this trend was the plot constructed for chromosome 1B (Appendix A), wherein samples from Kazakhstan and Tajikistan were grouped together, and these were separate from samples from Kyrgyzstan and Uzbekistan. Interestingly, PC1 revealed more differences between the accessions from Kazakhstan and Uzbekistan, as seven plots (1A, 2B, 2D, 3A, 3D, 5A, and 7D) showed distinctions between samples from these groups. The plots of chromosomes 2A, 2D, and 6A, using both PC1 and PC2, were informative. In general, these plots illustrate that the SNP markers in these chromosomes help determine regional adaptation. Therefore, the sets of SNPs on these chromosomes were further analyzed to aid in the discrimination between samples from within the Central Asian region via the selection of markers with different frequencies of SNP alleles. The spread of accessions in all 21 chromosomal PCoA plots suggests that the samples from Tajikistan were more closely related to samples from Afghanistan and samples from Kazakhstan were closer to samples from the Russian Federation (Appendix A).

### 2.4. The Selection of SNP Markers Associated with the Regional Adaptation of Winter Wheat in Central Asia

The evaluation of SNP markers with opposing frequencies of alleles within accessions of the Central Asian countries yielded 1006 informative SNPs that may play a role in plant adaption at the regional level (Appendix A). The assessment of these SNPs in the four neighboring countries in Central Asia suggested that 314, 237, 119, and 66 were differentiated in Kazakhstan, Uzbekistan, Tajikistan, and Kyrgyzstan, respectively. When the samples from the four countries were assessed in pairs, Tajikistan and Uzbekistan were the most strongly distinguished from the other two countries based on 81 SNPs, followed by Tajikistan and Kyrgyzstan, which were differentiated using 74 SNPs (Appendix A), confirming the genetic similarities in these countries. This list of 1006 SNPs with frequencies of opposite allele presence of more than 50% was also analyzed in relation to two additional case studies: (1) SNPs with opposing allele frequencies in the samples taken from Russia (northern neighbor) and Central Asian countries (Appendix A); (2) SNPs with opposing allele frequencies in samples taken from Afghanistan (southern neighbor) and Central Asian countries (Appendix A). In the first case, when the samples from the northern neighboring country were compared to the samples from Central Asia, Russia had the most SNPs in common with Kazakhstan (149 SNPs), followed by Uzbekistan (42 SNPs) and Tajikistan (12 SNPs). In the second case, when the samples from the southern neighboring country were compared to the samples from Central Asia, it was revealed that Afghanistan had the most SNPs in common with Uzbekistan (132 SNPs), followed by Tajikistan (52 SNPs) and Kazakhstan (24 SNPs).

As a number of PCoA plots constructed using sets of SNPs from individual chromosomes (Appendix A) showed differences in the grouping of accessions from Central Asia, the 1006 SNPs that were used for this assessment (Appendix A) were searched for genes that might be associated with regional adaptation in winter wheat. The assessment of the Wheat Ensembl database [26] suggested that out of 1006 SNPs, 555 were components of these genes (Appendix A). The evaluation of these 555 genes indicated both adaptability- and stress-resistance-associated genes. This list included genes for the F-box proteins that regulate photomorphogenesis, the circadian clock, and flowering time [27,28], which were located on chromosome 5A (*TraesCS5A02G208500* and *TraesCS5A02G208600*) and on chromosome 6A (gene *TraesCS6A02G407500*) (Appendix A). The *ZEITLUPE* (*ZTL*, *TraesCS6B02G149800*) gene is involved in the regulation of the circadian clock and flowering time [29]. Plant-development- and stress-resistance-associated genes represent genetic factors that are associated with ubiquitination, including the E3 ligase, which catalyzes the final step of the ubiquitination cascade [30]. In the above-mentioned list, E3 ligase was identified on chromosomes 1A, 2B, 2D, 3A, 5A, 6A, 6B, and 6D (Appendix A). In addition, the list also included other plant-growth- and stress-resistance-associated genes, including *WRKY* on chromosome 2B (*TraesCS2B02G552800*) and *ERD* (*EARLY RESPONSIVE TO DEHYDRATION*, *TraesCS2D02G129500*) on chromosome 2D (Appendix A).

## 3. Materials and Methods

### 3.1. Winter Wheat Collection

The winter wheat collection used in the current study consisted of 667 accessions from Europe (EU, 440), Central Asia (CA, 115), Afghanistan (AFG, 32), Russia (RUS, 26), Turkey (TUR, 19), the USA (19), and Australia (AUS, 16). The Central Asia collection itself is composed of 115 accessions, including accessions from Kazakhstan (KAZ, 52), Kyrgyzstan (KGS, 27), Uzbekistan (UZB, 27), and Tajikistan (TAJ, 9) (Appendix A). The collection was formed via the activities of CAWBIN, with the active participation of scientists from the UK, Turkey (CIMMYT), and Kazakhstan. The seeds of the collection were grown under greenhouse conditions at the John Innes Centre, with the application of a single seed descent (SSD) approach [31].

### 3.2. Genotyping Collection

A set of 192 winter wheat accessions of the 667 accessions in the CAWBIN panel were genotyped using a 35K Affymetrix Axiom SNP array [23] in this study. The genetic data for the remaining 475 accessions from Europe, the USA, and Australia were obtained from public databases [32]. Then, the two data sets were merged and further analyzed as described below.

### 3.3. Statistics Analysis

In total, 10,746 polymorphic SNP markers were selected for phylogenetic analysis using the previously published criteria to identify informative SNPs [33]. According to these criteria, markers with a call rate of >90%, a Hardy–Weinberg equilibrium fit of *p* > 0.001, a confidence score of 0.5, and a minor allele frequency (MAF) of >5% were considered to meet the requirements. An analysis of the population structure of the collection was performed using the Bayesian Markov Chain Monte Carlo (MCMC) algorithm in STRUCTURE [34]. K values of 2 to 10 were tested, the burn-in period was set to 100,000, and the number of MCMC replications after each burn-in was set to 100,000. A selection of the optimal number of subpopulations was made using the “Evanno test” [35] and the “elbow method” conceptualized by R. Thorndike [36]. A principal coordinate analysis (PCoA) based on Nei’s unbiased genetic distance, the number of effective alleles (Ne), Nei’s unbiased diversity (uh), and the percentage of polymorphic loci (% P) was undertaken using GenAlEx 6.5 [37].

## 4. Discussion

### 4.1. Comparative Population Structures of Spring and Winter Types of Wheat in Central Asia

It has previously been established that historic wheat development in Central Asia was based on two primary gene flow directions [13,14,15,17]. The first direction originated from the Northern European side, mainly via Russia and Ukraine, as these countries, along with some Central Asian republics, were part of the USSR. The second direction originated from the southwest, part of the ancient Silk Road [14,15], and has a lengthy trade history with its southern neighbors, including Afghanistan. This is particularly true for Kazakhstan, as this country’s wheat-growing region stretches from its northern borders with Russia to its southern borders with Kyrgyzstan and Uzbekistan. Given this country’s geographic and climatic differences, spring wheat cultivation dominates in its northern territories, and winter wheat growth prevails in its southern and southeastern territories. Recent reports on the genetic variation in global wheat, including in spring wheat from Kazakhstan, have indicated a strong genetic relationship between European and Kazakh wheat accessions [25]. Interestingly, by using 690 wheat samples that were genotyped by 3541 SNP markers, the authors showed a strong relationship between modern wheat accessions from the USA and Kazakhstan. This can be explained by the foundational role played by the Turkey Red Wheat (TRW) type in the development of modern wheat varieties in the USA [38], as TRW was initially brought to the USA by Russian Mennonites (ethnically German) from 1874 [38,39]. Furthermore, another study found that genetic germplasms from the territories of modern Ukraine and the Russian Federation were successfully introduced into northern Kazakhstan in the middle of the last century [40]. However, this study did not address the winter type, from either Kazakhstan or the other countries of Central Asia, and thus cannot provide a complete picture of the genetic structures of both types of wheat in the region. Therefore, this work assessed a collection of 667 winter wheat samples from different parts of the world, including 115 accessions from Central Asia, using 10,746 polymorphic SNPs. The results regarding genetic structure, derived using the STRUCTURE package at step K4, suggest that the samples from Central Asia can be split into two groups, cluster 1 and cluster 3 (Figure 3). Interestingly, all samples from Afghanistan were grouped in cluster 1, while the majority of accessions from Russia mostly fit into cluster 3, suggesting that two different genetic pools contributed to the genetic diversity of samples from Central Asia. The evaluation of the samples in these two K4 clusters showed that accessions from Tajikistan and Uzbekistan were also common with samples Afghanistan, while the majority of samples from Kazakhstan and Kyrgyzstan were grouped together with samples from Russia (Figure 3). The analyses of the clusters in steps K5–10 also confirmed this trend, suggesting that breeders from Kazakhstan and Kyrgyzstan have actively used the genetic germplasm from the Russian Federation, and those from Tajikistan and Uzbekistan have successfully utilized genetic resources of wheat from regions neighboring to the southwest. Despite this, in our analyses of the PCoA plots of the groups in the Central Asian samples (Figure 1), PC2 separated the groups of accessions from Kazakhstan from the groups from the remaining three countries. Therefore, since these three countries share similar geographic and climatic conditions suitable for winter wheat growth, this result may reflect active germplasm exchange among breeding communities in Kyrgyzstan, Tajikistan, and Uzbekistan.

As wheat was initially domesticated in the Middle East [41], the samples in our collection were compared to a group of modern cultivars from Turkey, which were the only available representatives from that region. The PCoA plots showed that the eigenvalues of those samples from Turkey were in the middle of those from the other countries, which agrees closely with the broadly accepted conceptualization of wheat evolution [42]. Notably, the accessions from Kyrgyzstan (0.010), Uzbekistan (0.10), and Tajikistan (0.012) were genetically closest to samples from Turkey, followed by Russia (0.021) and Kazakhstan (0.026) (Figure 1, Table 2).

### 4.2. Patterns of PCoA Plots Using SNPs on Individual Wheat Chromosomes

Our previous study seeking to identify the genetic structure within a barley collection suggested that evaluation of the PCoA plots of groups of SNP markers in individual chromosomes may provide additional insights into the identification of genes associated with plant adaptation [43]. As such, in this study, we applied a similar approach to the search for genetic factors contributing to plant adaptability in the Central Asian winter wheat collection and assessed groups of accessions using SNPs on 21 wheat chromosomes. This approach is particularly suitable to the identification of SNPs on an intraregional level, i.e., in Central Asia, where the samples have a high level of genetic identity (Table 1). Naturally, the analyses of 21 PCoA plots showed different degrees of closeness between the studied groups (Appendix A). When the locations of samples from Kazakhstan and Tajikistan (the two most genetically distant groups within the region (Table 1)) were compared to identify their sets of accessions, the most informative plots were for chromosomes 2A, 5A, 6A, 6D, and 7A, constructed using PC1, and for 1D, 2A, 3B, 4B, and 7B, constructed using PC2, respectively (Appendix A). PC1 was also useful when used for the differentiation of accessions between Kazakhstan and Uzbekistan via the PCoA plots of chromosomes 1A, 2B, 2D, 3A, 3D, 5A, and 7D (Appendix A). Therefore, we can conclude that the SNPs on these chromosomes may be useful markers when searching for genes associated with plant adaption in Central Asia.

### 4.3. Identification of Genes Associated with Plant Adaptation in the Central Asian Region

One of the most logical approaches to identifying genes associated with plant adaptation at both intra- and interregional levels is the evaluation of SNPs with opposing allele frequencies in samples with different origins [44]. In this work, the assessment of samples from Central Asian countries using 10,746 polymorphic SNPs allowed for the extraction of 1006 markers with opposite allele frequencies in at least one of the four groups of Central Asian accessions (allele frequencies of 50% or higher). As we expected, the region with the largest number of SNPs identified with opposing allele frequencies in Central Asia was Kazakhstan (314 SNPs), followed by Uzbekistan (237), Tajikistan (119), and Kyrgyzstan (66) (Appendix A). When the samples from the four countries were analyzed in pairs of groups, the largest sets of SNPs were associated with Tajikistan and Uzbekistan (81), followed by Tajikistan and Kyrgyzstan (74) (Appendix A), confirming the genetic closeness amongst these three countries. The assessment of these samples with Russian and Afghan accessions further clarified the genetic compositions of the Central Asian groups of wheat. It was shown that samples from Russia shared the most SNP alleles with Kazakhstan (149 SNPs), followed by Uzbekistan (42) and Tajikistan (12). In contrast, the samples from Afghanistan showed similar SNP allele frequencies to those from Uzbekistan (132 SNPs), followed by Tajikistan (52) and Kazakhstan (24). Therefore, it was concluded that Kazakhstan’s historic winter wheat breeding process was primarily developed using germplasm from Russia. This assumption agrees well with previous reports on the history of the development of wheat breeding in Kazakhstan. In particular, Udachin and Shahmedov (1984) noted that the country’s wheat breeding development changed drastically during the Soviet era; this change was heavily influenced by cooperation with Russian breeders. On the contrary, breeding activities in Uzbekistan and Tajikistan were expanded primarily with the utilization of germplasms from countries neighboring to the southwest, including Afghanistan [12]. Additionally, after the breakup of the USSR, breeders from Central Asian countries began to collaborate closely via bilateral and international projects [2,13]. This is clear in the case of Kyrgyzstan, as the results from this study and previously published reports indicate that breeders in this country have extensively exchanged wheat germplasm with the other three Central Asian countries [45].

A further study of the 1006 SNPs with opposite allele frequencies identified in the four groups of accessions from Central Asia revealed that 555 are components of the genes listed in the Wheat Ensembl database (Appendix A). A cursory evaluation of these genes suggests they are primarily associated with plant stress responses. In particular, there is a notable presence of genes related to the ubiquitination of proteins (Appendix A). Ubiquitination refers to the well-examined post-translational regulation of various biological processes, including growth and development, responses to biotic and abiotic stresses, and the regulation of the chromatin structure [46,47]. In this study, at least nine of the genes participating in the conjugation and ligation steps of ubiquitination were present with opposite allele frequencies (Appendix A). The other genes that feature in the SNP table are associated with stress resistance, including *WRKY* and *ERD,* located on chromosomes 2B and 2D, respectively (Appendix A). As expected, this list includes a number of flowering-time-associated genes, as this property is important to plants’ adaptations to different environments. One such gene is *ZTL* (chromosome 6B), which controls the circadian clock and flowering time [29]. Other examples include SNPs that are part of the F-box proteins involved in photomorphogenesis, circadian clock regulation, and the control of flowering time [27,28]. Thus, identifying these adaptation-related genes may be beneficial to regional winter wheat breeding projects and the development of new competitive commercial cultivars in Central Asia.

## 5. Conclusions

This evaluation of winter wheat samples collected from different parts of the world using 10,746 polymorphic SNPs suggests that the range of wheat genetic diversity within Central Asian countries is similar to that in other studied regions. Within the Central Asian region, Nei’s genetic diversity index values of accessions ranged from 0.212 in Kazakhstan to 0.282 in Uzbekistan, and these values are comparable to those in Russia (0.258) and Afghanistan (0.259). The PCoA plots constructed using 10,746 SNPs showed the genetic identities of samples from four Central Asian countries, which differed from those from two neighboring countries. Unlike Kazakhstan, where samples showed genetic similarities to samples from Russia (0.02), the accessions from Kyrgyzstan (0.01), Tajikistan (0.012), and Uzbekistan (0.01) were genetically close to samples from Turkey. Furthermore, assessment using “STRUCTURE” confirmed that, at step K4 (one of the most important steps in the differentiation of samples), the accessions from Uzbekistan and Tajikistan grouped together with samples from their southern neighbor, Afghanistan, while accessions from Kazakhstan and Kyrgyzstan grouped with samples from their northern neighbor, the Russian Federation. These results suggest that historic breeding projects in Kyrgyzstan have actively used genetic resources of winter wheat from Kazakhstan and Russia. A comparative evaluation of the samples from the four Central Asian countries, using 10,746 SNPs, allowed us to identify 1006 SNPs with opposite allele frequencies, suggesting that these markers may be associated with plant adaptation traits at the intraregional level. The results of this study illustrate the genetic structures of accessions with different origins and suggest the possible directions of gene flow in the winter wheat accessions from the four countries in Central Asia. In addition, we identified a set of SNPs showing opposite allele frequencies in the four groups of samples from Central Asia. This set can be further utilized to search for candidate genes relevant to regional winter wheat breeding activities.

## Figures and Tables

**Figure 1 plants-12-02233-f001:**
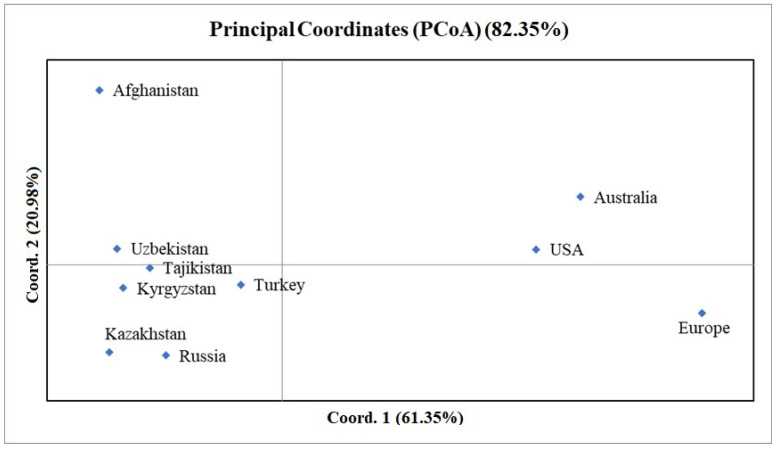
Principal coordinate analysis based on Nei’s unbiased genetic distance for the 667 accessions of the winter wheat collection using the SNP genotyping results.

**Figure 2 plants-12-02233-f002:**
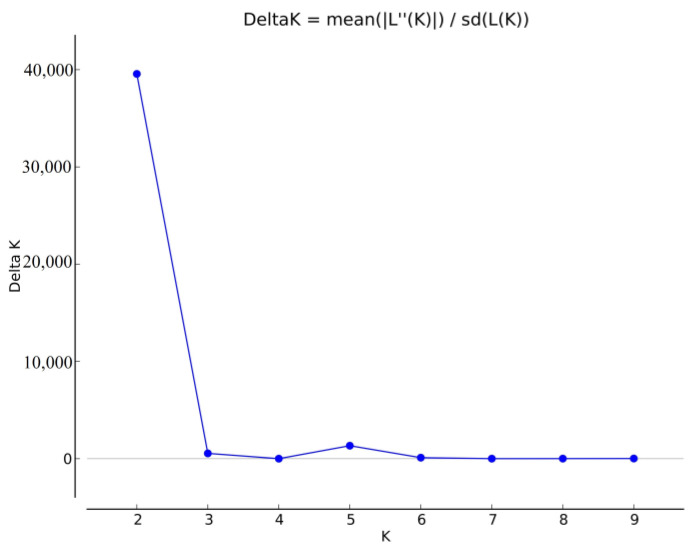
Evanno evaluation of the optimal number within the winter wheat population using the delta K value based on the “Evanno” and “elbow” methods.

**Figure 3 plants-12-02233-f003:**
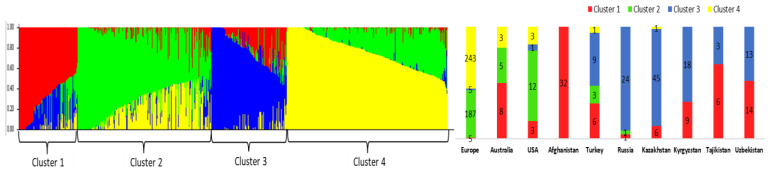
Graphical representation (K4) of the clusters of winter wheat accessions using the STRUCTURE package.

**Table 1 plants-12-02233-t001:** Genetic diversity of 667 common winter wheat accessions using 10,746 polymorphic SNP markers.

Groups	n	Ne	uh	% P
Europe	440	1.486 ± 0.003	0.292 ± 0.002	99.26%
Australia	16	1.468 ± 0.003	0.297 ± 0.002	82.68%
USA	19	1.516 ± 0.003	0.325 ± 0.002	87.45%
Afghanistan	32	1.420 ± 0.003	0.259 ± 0.002	82.87%
Turkey	19	1.484 ± 0.003	0.308 ± 0.002	88.88%
Russia	26	1.406 ± 0.003	0.258 ± 0.002	86.09%
Kazakhstan	52	1.325 ± 0.003	0.212 ± 0.002	86.73%
Kyrgyzstan	27	1.434 ± 0.003	0.270 ± 0.002	83.08%
Tajikistan	9	1.418 ± 0.003	0.280 ± 0.002	69.71%
Uzbekistan	27	1.460 ± 0.003	0.282 ± 0.002	87.78%
Total	667	1.428 ± 0.001	0.267 ± 0.001	84.94 ± 2.31%

Note: n = No of lines; Ne = No. of effective alleles; uh = Nei’s unbiased diversity index; % P = percentage of polymorphic loci.

**Table 2 plants-12-02233-t002:** Nei’s genetic distances among ten winter wheat groups with different origins.

Origins	EU	AUS	USA	AFG	TUR	RUS	KAZ	KGS	TAJ
AUS	0.084								
USA	0.064	0.041							
AFG	0.280	0.183	0.168						
TUR	0.131	0.091	0.067	0.082					
RUS	0.180	0.154	0.111	0.127	0.021				
KAZ	0.216	0.181	0.134	0.122	0.026	0.020			
KGS	0.197	0.142	0.111	0.073	0.010	0.018	0.015		
TAJ	0.196	0.129	0.104	0.077	0.012	0.026	0.042	0.012	
UZB	0.207	0.135	0.111	0.050	0.010	0.027	0.028	0.003	0.007

Note: EU—Europe. AUS—Australia. AFG—Afghanistan. TUR—Turkey. RUS—Russia. KAZ—Kazakhstan. KGS—Kyrgyzstan. TAJ—Tajikistan.

## Data Availability

Publicly available datasets that analyzed in this study can be found here: [https://www.cerealsdb.uk.net/cerealgenomics/CerealsDB/indexNEW.php]. In addition, the data presented in this study are available on request from the corresponding author.

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
