# Peer review of "Population Structure of Modern Winter Wheat Accessions from Central Asia"

_plants, 2023, doi:10.3390/plants12122233_

Round 1
Reviewer 1 Report
see my two comments in attached pdf

Author Response
Replies to comments and suggestions of Reviewer 1.
Question 1. Which chromosome had least number of SNP like wise which of three genome had minimum SNP and why.
Reply: Thank you for your question. The least number of SNP markers was registered in chromosome 6D (115 SNPs). The minimum number of markers among the three genomes was in the D genome (1526 SNPs). The survey of scientific publications in the literature suggests that the D genome has the smallest number of SNP markers in genotyping arrays.
To clarify this point, we added lines 212-216 in the section Results.
The most obvious reason for the differences between the three genomes is the recent evolution of the D genome (1-2 million years ago) when compared to A and B genomes (about 7 million years ago) (Muqaddasi et al., 2017).
Since it is well known fact, and there is no novelty on this issue, we did not discuss it in the section Discussion.
Question 2: Was authors found any effect of number of SNP (chromosome wise) on PCoA values
Reply: Thank you for your question. In our opinion, the differences in the number of SNP markers per chromosome did not affect the outputs of the PCoA analyses. For instance, when PCoA plots for 1B (most numbers of SNPs) and 6D (least number of SNPs) were compared, in both plots, 1) the closest group for Turkey samples were samples from Tajikistan, 2) samples from Afghanistan were significantly far from other groups, and 3) samples from Russia were distinctly different from samples of Central Asia. Also, the least number of SNP markers per chromosome was 115 SNPs (chromosome 6D). Obviously, it is a large enough number of markers for evaluating chromosome specificity on relationships among taxa. Therefore, we decided that we should not stress on differences in SNP numbers per chromosome in the manuscript.

Reviewer 2 Report
Central Asian countries are important wheat producer with unique climate and ecological conditions. The wheat cultivars in those countries possess valuable traits and gene resources that adapted to the local soil, cultivation, and stress conditions. However, very little is known for the wheat cultivars in those countries regarding to their genetic diversity, structure and gene flow with cultivars from other countries. This paper reported the population structure analysis of 115 modern winter wheat cultivars from four four Central Asian countries (Kazakhstan, Kyrgyzstan, Tajikistan and Uzbekistan) and compared with germplasms from six other geographic origins using polymorphic single-nucleotide polymorphism (SNP) markers. The genetic relationship of wheat cultivars from those countries to nearby countries (Russia, Afghanistan, Turkey), as well as Europe, Australia and USA were studied. The information is very valuable to wheat researchers and breeders not only in the central Asian countries, but also to worldwide wheat societies.
I have one concerns regarding to the current manuscript. The authors studied the population structure of mordern winter wheat cultivars. Do you have relative information of local wheat landraces to be involved in the SNP data analysis? It would be better to see the historical breeding process in shaping the genetics structure of the current mordern cultivars.
Author Response
Replies to comments and suggestions of Reviewer 2.
Comment 1: I have one concerns regarding to the current manuscript. The authors studied the population structure of modern winter wheat cultivars. Do you have relative information of local wheat landraces to be involved in the SNP data analysis? It would be better to see the historical breeding process in shaping the genetics structure of the current mordern cultivars.
Reply: Thank you for your comment. Unfortunately, our records showed none of the landraces' seeds in existing collections of winter wheat in Central Asian seed banks. The oldest available cultivar was Bezostaya 56, which was released in the middle of the last century. However, Bezostaya 56 was released by Russian breeders in the USSR and technically can not be considered as the landrace. Therefore, our study was focused only on modern genotypes of these Central Asian countries.

Reviewer 3 Report
The manuscript „Population structure of modern winter wheat accessions from Central Asia” by Amalova et al. offers insights on population structure of wheat accessions from Kazakhstan, Kyrgyzstan, Uzbekistan and Tajikistan and brings out some valuable information to people who are actually working in the field of wheat breeding. Manuscript is interesting and well written. A large amount of work was involved in the study.
Before publishing it requires some really minor revisions.
Please, check the values of Afghanistan – Uzbekistan, Tajikistan and Kazakhstan common SNPs (lines 255, 256, 365, 366). They are not congruent with the values show in Table S5 (Uzbekistan 132 SNPs, Tajikistan 52 SNPs and Kazakhstan 24 SNPs). In addition, please delete Table S4 in parenthesis (lines 266 and 274). The genes are listed in Table S6.
Author Response
Replies to comments and suggestions of Reviewer 3.
Suggestion 1: Please, check the values of Afghanistan – Uzbekistan, Tajikistan and Kazakhstan common SNPs (lines 255, 256, 365, 366). They are not congruent with the values show in Table S5 (Uzbekistan 132 SNPs, Tajikistan 52 SNPs and Kazakhstan 24 SNPs). In addition, please delete Table S4 in parenthesis (lines 266 and 274). The genes are listed in Table S6.
Reply: Thank you for your valuable suggestion! We are sorry for this typo. We corrected the sentences about the number of common SNPs (lines 259-260, 369-370). Also, we corrected the tables from TableS4 to TableS6 (lines 270, 27).
